# Effects of Wearing Face Masks on Cardiorespiratory Parameters at Rest and after Performing the Six-Minute Walk Test in Older Adults

**DOI:** 10.3390/geriatrics7030062

**Published:** 2022-06-07

**Authors:** Patchareeya Amput, Sirima Wongphon

**Affiliations:** 1Department of Physical Therapy, School of Allied Health Sciences, University of Phayao, Phayao 56000, Thailand; 2Unit of Excellent of Physical Fitness and Exercise, University of Phayao, Phayao 56000, Thailand; 3Department of Traditional Chinese Medicine, School of Public Health, University of Phayao, Phayao 56000, Thailand; sirima.wo@up.ac.th

**Keywords:** COVID-19, face masks, 6 min walk test, cardiorespiratory fitness, older adults

## Abstract

The effects of wearing cloth masks and surgical masks were investigated on respiratory rate, heart rate, blood pressure, oxygen saturation and perceived exertion at rest and after performing a six-minute walk test (6MWT) in older adults. Forty older adults were recruited and randomized into six groups including wearing no mask, cloth masks and surgical masks, at rest and during the 6MWT. At rest, all subjects sat quietly wearing no mask, a cloth mask or a surgical mask. All subjects performed a 6MWT by walking as fast as possible without running while wearing no mask, a cloth mask or a surgical mask. Respiratory rate, heart rate, blood pressure, oxygen saturation and perceived exertion were assessed before and after the rest and the 6MWT. Results showed that cloth masks and surgical masks did not impact cardiorespiratory parameters at rest or after performing a 6MWT, while an increase in perceived exertion was apparent in the groups wearing surgical masks and cloth masks after performing the 6MWT (*p* < 0.01). Cloth masks and surgical masks did not have an impact on cardiorespiratory fitness at rest and after performing the 6MWT in older adults.

## 1. Introduction

The coronavirus disease 2019 (COVID-19) first appeared in Wuhan, China and then spread rapidly worldwide [1]. COVID-19 causes serious medical problems such as colds, croup and bronchiolitis [2]. Coronavirus is transmitted from person to person by respiratory droplets [3,4]. Coronavirus transmission is reduced by social distancing, hand washing, wearing a mask and lockdown [5]. Previous reports recommended wearing masks to decline the spread of the disease [6,7]. Various kinds of masks are available including cloth masks, surgical masks and N95 masks, with the type of mask chosen depending on individual preference [5]. Furthermore, De-Yñigo-Mojado B et al. [8] found that surgical masks showed no statistically significant difference in terms of fit factor scores between healthcare providers with and without facial hair. However, the fit factor scores were statistically significantly different between the two groups, indicating that the fit factor effectiveness of filtering respirators was reduced in healthcare providers with facial hair. These findings suggested that filtering respirators provide a better fit factor than surgical masks. Older adults have an increased risk from COVID-19, with comorbidities and immunosenescence that stimulated viral-activated cytokine storms, leading to respiratory failure and multi-system involvement [9]. Thus, treatment of COVID-19 in older adults is more complex [9]. Consequently, the protection of older adults from coronavirus is crucial. Previous reports recommended that older adults should wear surgical masks and cloth masks [5,10]. However, some people find mask-wearing uncomfortable, with impaired gas exchanges [11,12].

Several studies found that wearing a mask during physical activities increased ventilation and also increased breathing resistance [13,14]. The World Health Organization (WHO) advises that wearing a mask during vigorous physical activities increases breathing difficulties [15], with reduced oxygen uptake and increased carbon dioxide rebreathing [15]. Reychler G et al. [16] found that healthy adults wearing a surgical mask had no influence on dyspnea, respiratory rate, heart rate, oxygen saturation and exercise performance during a short submaximal exercise test. Healthy volunteers wearing surgical masks experienced higher dyspnea than those without masks, but this did not impact results after a 6MWT [12]. However, the effects of wearing cloth masks and surgical masks on cardiorespiratory parameters at rest and after a submaximal exercise test using a 6MWT in older adults have not been evaluated. More research is needed to identify whether wearing cloth masks or surgical masks significantly affects performance measures when resting and completing the 6MWT in older adults. Therefore, this study aims to evaluate the effects of wearing cloth masks and surgical masks on respiratory rate, heart rate, blood pressure, oxygen saturation and perceived exertion at rest and after performing a 6MWT in older adults. We tested the hypothesis that cloth masks and surgical masks have more impact on cardiorespiratory parameters than no mask.

## 2. Materials and Methods

### 2.1. Study Design

A randomized crossover trial was used to investigate the effects of wearing cloth masks and surgical masks on respiratory rate, heart rate, blood pressure, oxygen saturation and perceived exertion at rest and after performing a 6MWT in older adults.

### 2.2. Participants

Forty older adults living in Phayao Province, Thailand, voluntarily participated to take part in this study. Subjects were randomly assigned to six conditions with forty subjects per condition, including: (1) sitting quietly wearing no mask for 10 min, (2) sitting quietly wearing a cloth mask for 10 min, (3) sitting quietly wearing a surgical mask for 10 min, (4) performing a 6MWT with no mask, (5) performing a 6MWT wearing a cloth mask and (6) performing a 6MWT wearing a surgical mask. The sample size was calculated using a power of 0.8, power analysis with an effect size f of 0.20 and alpha of 0.05 [17]. Inclusion and exclusion criteria were as follows: aged 65 or older with normal body mass index (BMI = 18.5–24.9 kg/m^2^) [18] and can perform a 6MWT, no previous problems of unstable cardiorespiratory diseases, neurological diseases with imbalance problems, musculoskeletal disease affecting standing or walking and arterial hypertension with the possibility of left ventricular dysfunction was measured using an echocardiogram. This study was approved by the Clinical Research Ethics Committee of the University of Phayao, Phayao, Thailand (1.3/066/64).

### 2.3. Testing Procedure

The subjects were interviewed and assessed on baseline demographic data including height, weight, BMI, respiratory rate, heart rate, blood pressure, oxygen saturation and underlying diseases. A CONSORT statement was used in this study. Each subject performed six protocols, including: (1) sitting quietly wearing no mask for 10 min, (2) sitting quietly wearing a cloth mask for 10 min, (3) sitting quietly wearing a surgical mask for 10 min, (4) performing a 6MWT with no mask, (5) performing a 6MWT wearing a cloth mask and (6) performing a 6MWT wearing a surgical mask. Test sequences were randomly assigned using the website randomizer.org. Respiratory rate, heart rate, blood pressure, oxygen saturation and perceived exertion were measured before and after each protocol, while distance covered during the 6MWT was measured at the end of each test.

The cloth masks were 3-layer ear-loop handmade masks. The 3 layers were made of muslin fabric, while the surgical masks used were standard 3-ply disposable face masks (SKIN US LOC Co., Ltd., Nonthaburi, Thailand). The masks were tightly fitted to the nose and face of the subjects.

For the 6MWT, 2 reflective cones indicated the turnaround points. Reflective tape was located at the start and finish lines and at every 3 m of the walking course. The time was monitored using a stopwatch. Before test, the older adults were asked to wear comfortable clothes and shoes. They were instructed not to wear a mask and to sit on a chair with a normal breathing for 30 min. Each older adult performed the 6MWT on an individual basis. The first researcher encouraged the older adults to walk as fast as possible without running for the six minutes of the test duration, continuing at the same pace without stopping. The distance completed in each 6MWT was recorded [19]. After completing the 6MWT, the subjects were instructed to wear their masks and proceed to the assessment area. The second researcher did not know to which group the subjects had been assigned and had no entry to the registration number or the area of the 6MWT test. Respiratory rate, heart rate, blood pressure, oxygen saturation and perceived exertion were assessed before and after the 6MWT. At 1 min after the 6MWT, heart rate and pulse oxygen saturation were measured with a finger pulse oximeter (SB200, Rossmax International Ltd., Taipei, Taiwan). Blood pressure was measured using an automatic blood pressure monitor (HEM-8712, Omron healthcare Co., Ltd., Kyoto, Japan). Respiratory rate was investigated by observation, watching the rise and fall of the chest of each subject and counting the number of times that they complete a breath cycle including inhalation and exhalation within a fixed period of time to define the number of breaths per minute [20]. Borg rating of perceived exertion scale (6–20 scales) was used to evaluate perceived exertion. Sixty minutes of rest were allocated between each 6MWT to minimize fatigue [18]. A flow diagram of the participants through each stage of this study is shown in Figure 1.

### 2.4. Statistical Analysis

Normal distribution was calculated using the Shapiro–Wilk test. Subject demographics were analyzed using descriptive statistics, with a two-way repeated measures ANOVA test used to compare respiratory rate, heart rate, blood pressure, oxygen saturation, perceived exertion and walking distance during the 6MWT both at rest and after performing the 6MWT. Bonferroni post hoc tests were used to assess pairwise comparisons and significant F ratios. All statistical analyses were conducted using IBM SPSS Statistics software, version 22.0 (IBM Corp., Armonk, NY, USA), with a *p*-value of less than 0.05 set to denote significance.

## 3. Results

Forty older adults voluntarily participated in this study with demographics shown in Table 1. Baselines of respiratory rate, heart rate, blood pressure and oxygen saturation were normal. Results indicated that comorbidities included hypertension, diabetes mellitus and dyslipidemia.

All older adults completed the three conditions of face masks when sitting quietly. No differences in respiratory rate, heart rate, blood pressure and oxygen saturation were recorded among the conditions, with results presented in Table 2. Wearing cloth masks and surgical masks slightly increased the respiratory rate, heart rate and blood pressure after sitting quietly for 10 min.

All older adults completed the three conditions of the 6MWT. No differences in respiratory rate, heart rate, blood pressure and oxygen saturation were recorded among the groups, with results presented in Table 3. Wearing cloth masks and surgical masks significantly increased rated perceived exertion (RPE) scores for perceived exertion when compared to no mask (*p* < 0.01). The 6MWT distances were not significantly different among the groups. 

The six face mask protocols showed no differences in respiratory rate, heart rate, blood pressure and oxygen saturation among the groups. Performing the 6MWT with no mask, a cloth mask and a surgical mask significantly increased perceived exertion compared to sitting quietly with no mask, a surgical mask and a cloth mask (*p* < 0.01), with results presented in Table 4.

## 4. Discussion

The study findings determined that cloth masks and surgical masks had no impact on cardiorespiratory parameters including respiratory rate, heart rate, blood pressure, oxygen saturation at rest and after performing the 6MWT, while wearing face masks impacted perceived exertion after the 6MWT.

The cloth masks and surgical masks did not impact respiratory rate, heart rate, blood pressure and oxygen saturation after sitting for 10 min. These findings concurred with a previous study [21]. The authors tested cloth masks and surgical masks in 50 hospital employees (median age 33 years; 32% with comorbidities) while sitting for 10 min. The results showed that sitting while wearing a cloth mask and a surgical mask brought about no statistically significant differences in heart rate and oxygen saturation when compared to the baseline. In addition, another previous study investigated surgical masks at 5 and 30 min of rest in 15 healthy volunteers (average age 31.1 ± 1.9 years) and 15 veterans with severe chronic obstructive pulmonary disease (COPD) (average age 71.6 ± 8.7 years). Those results found no major change in oxygen saturation at 5 and 30 min of rest in healthy volunteers and severe COPD. Furthermore, the authors did not compare oxygen saturation between two populations [22]. These findings suggested that cloth masks and surgical masks did not impact the respiratory rate, heart rate, blood pressure and oxygen saturation at rest for less than 30 min in individuals with or without underlying pulmonary disease [22]. Our study tested the three conditions of face masks in 40 older adults without pulmonary disease sitting for 10 min. The results showed that no differences in respiratory rate, heart rate, blood pressure and oxygen saturation were recorded among the conditions. Therefore, further studies may consider longer time periods and compare cardiorespiratory parameters between healthy volunteers and those with pulmonary disease.

After performing the 6MWT, respiratory rate, heart rate and blood pressure slightly increased, while oxygen saturation slightly decreased. This change occurred because the prolonged inspiratory activity leads to high negative intrathoracic pressure, leading to increased cardiac preload, systolic volume and cardiac afterload, respectively [23,24]. This gave an increased respiratory rate, heart rate and blood pressure. Decreased oxygen saturation reflected reduced oxygen levels in the blood. These results concurred with previous studies showing that healthy subjects with no mask and surgical masks had no statistically significant differences in respiratory rate, heart rate and oxygen saturation after performing the 6MWT [12,25]. In contrast, our results found that wearing cloth masks and surgical masks statistically significantly increased perceived exertion when compared with no mask. The face masks increased airflow resistance, facial skin temperature and moisture or heat of the inhaled air, resulting in increased dyspnea [26,27,28]. Previous studies reported that healthy subjects wearing face masks recorded similar distances walked during the 6MWT compared to no mask [17,25,29], while changes after the 6MWT were found in heart failure or chronic lung disease patients [30,31]. This study involved no participants with heart failure or chronic lung disease.

There were several limitations. First, in this study there was no comparison made between men and women for the different cardiorespiratory parameters and distance of the 6MWT in the six mask conditions. This matter of comparison of the effects of face masks on cardiorespiratory parameters between men and women should be investigated in further studies. Secondly, further studies should evaluate the psychological status of the volunteers, taking into consideration anxiety and stress, since these conditions can increase dyspnea when wearing a facemask, and can be a source of bias in the results.

## 5. Conclusions

Cloth masks and surgical masks had no impact on respiratory rate, heart rate, blood pressure, oxygen saturation at rest and after performing the 6MWT, but had an impact on perceived exertion after the 6MWT in older adults.

## Figures and Tables

**Figure 1 geriatrics-07-00062-f001:**
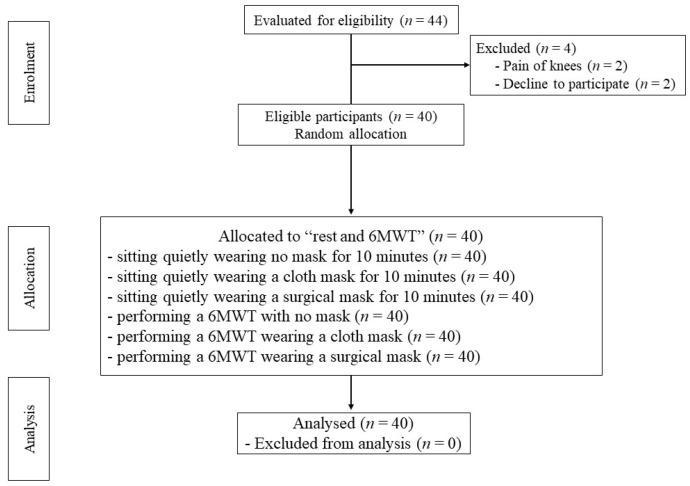
A flow diagram of the participants through each stage of this study.

**Table 1 geriatrics-07-00062-t001:** Characteristics of the subjects. Values are means ± SD.

Variables	*n* = 40 (F = 25, M = 15)
**Sex, *n* (%)**	
Female	25 (62.5)
Males	15 (37.5)
Age (years)	69.60 ± 3.46
Weight (kg)	51.03 ± 5.91
High (cm)	155.23 ± 8.47
BMI (kg/m^2^)	21.14 ± 1.26
Heart rate (bpm)	80.0 ± 10.86
Respiratory rate (breaths per minute)	16.5 ± 2.41
Systolic blood pressure (mmHg)	139.73 ± 14.99
Diastolic blood pressure (mmHg)	74.70 ± 7.97
SpO_2_ (%)	98 ± 1.05
**Comorbidities; *n* (%)**	
-None	17 (42.5)
-Hypertension	12 (30)
-Diabetes mellitus	6 (15)
-Dyslipidemia	5 (12.5)

Abbreviations: *n* = number; F = female; M = male; kg = kilograms; cm = centimeters; BMI = body mass index; bpm = beats per minute; % = percentage.

**Table 2 geriatrics-07-00062-t002:** Effects of face masks on cardiorespiratory parameters at rest.

Variables	Duration	No Mask	Cloth Mask	Surgical Mask	*p*-Value
Respiratory rate (breaths per minute)	Before sitting	17.2 ± 1.63	17.1 ± 1.91	17.3 ± 2.36	0.06
	After sitting	17.3 ± 1.97	19.0 ± 2.90	18.8 ± 2.98	
Heart rate (bpm)	Before sitting	75.9 ± 8.82	75.9 ± 7.93	75.2 ± 8.79	0.75
	After sitting	76.3 ± 8.28	77.7 ± 8.23	77.6 ± 5.55	
Systolic blood pressure (mmHg)	Before sitting	140.72 ± 14.13	138.75 ± 9.65	141.85 ± 7.38	0.85
	After sitting	141.20 ± 9.01	140.88 ± 7.00	142.88 ± 6.27	
Diastolic blood pressure (mmHg)	Before sitting	76.03 ± 7.50	76.73 ± 6.62	77.23 ± 7.55	0.10
	After sitting	76.53 ± 7.38	76.95 ± 6.52	77.55 ± 6.06	
SpO_2_ (%)	Before sitting	98.18 ± 0.81	98.40 ± 0.71	98.25 ± 0.71	0.38
	After sitting	98.10 ± 0.90	98.98 ± 0.73	97.98 ± 0.86	

Abbreviations: bpm = beats per minute; SpO_2_ = pulse oxygen saturation; % = percentage.

**Table 3 geriatrics-07-00062-t003:** Effects of face masks on cardiorespiratory parameters at 6MWT.

Variables	Duration	No Mask	Cloth Mask	Surgical Mask	*p*-Value
Respiratory rate (breaths per minute)	Pre-6MWT	19.0 ± 1.27	20.0 ± 32.48	20.0 ± 2.68	0.95
	Post-6MWT	22.2 ± 2.98	23.0 ± 4.10	23.1 ± 3.15	
Heart rate (bpm)	Pre-6MWT	75.9 ± 9.58	79.1 ± 11.35	80.2 ± 10.50	0.84
	Post-6MWT	81.6 ± 10.77	83.1 ± 12.02	84.1 ± 11.42	
Systolic blood pressure (mmHg)	Pre-6MWT	136.78 ± 10.37	132.90 ± 13.17	130.68 ± 13.63	0.37
	Post-6MWT	140.00 ± 10.43	140.20 ± 10.34	138.78 ± 11.92	
Diastolic blood pressure (mmHg)	Pre-6MWT	78.30 ± 7.56	78.90 ± 9.37	79.78 ± 8.87	0.76
	Post-6MWT	81.68 ± 8.92	80.48 ± 8.71	81.38 ± 9.23	
SpO_2_ (%)	Pre-6MWT	98.18 ± 0.64	98.03 ± 1.03	98.33 ± 0.62	0.57
	Post-6MWT	97.50 ± 1.09	97.68 ± 1.16	97.83 ± 1.15	
RPE	Post-6MWT	7.03 ± 1.21	8.25 ± 1.66	7.35 ± 1.23	<0.01
Distance (m)	Post-6MWT	390.65 ± 43.62	382.73 ± 49.560	383.80 ± 68.36	0.78

Abbreviations: bpm = beats per minute; SpO_2_ = pulse oxygen saturation; % = percentage; RPE = Rate of Perceived Exertion; m = meter.

**Table 4 geriatrics-07-00062-t004:** Effects of face masks on cardiorespiratory parameters at rest and 6MWT.

Variables	Duration	No Mask	Surgical Mask	Cloth Mask	*p*-Value
Heart rate (bpm)	Rest	76.33 ± 8.29	77.55 ± 5.55	77.73 ± 8.23	0.90
	6MWT	81.60 ± 10.77	84.10 ± 11.42	83.08 ± 12.02	
Respiratory rate (breaths per minute)	Rest	17.33 ± 1.97	18.78 ± 2.98	18.95 ± 2.90	0.65
	6MWT	22.25 ± 2.98	23.13 ± 3.15	22.98 ± 4.10	
Systolic blood pressure (mmHg)	Rest	141.20 ± 9.01	142.88 ± 6.27	140.87 ± 7.00	0.46
	6MWT	140.00 ± 10.43	138.78 ± 11.92	140.20 ± 10.34	
Diastolic blood pressure (mmHg)	Rest	76.53 ± 7.38	77.55 ± 6.06	76.95 ± 6.52	0.79
	6MWT	81.68 ± 8.92	81.38 ± 9.23	80.48 ± 8.71	
SpO_2_ (%)	Rest	98.10 ± 0.90	97.98 ± 0.86	97.98 ± 0.73	0.35
	6MWT	97.50 ± 1.09	97.83 ± 1.15	97.68 ± 1.16	
RPE	Rest	6.00 ± 0.00	6.15 ± 0.43	6.32 ± 0.62	0.01
	6MWT	7.02 ± 1.21	7.35 ± 1.23	8.25 ± 1.66	

Abbreviations: bpm = beats per minute; SpO_2_ = pulse oxygen saturation; % = percentage; RPE = Rate of Perceived Exertion.

## Data Availability

Not applicable.

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
