# Peer review of "Effects of Wearing Face Masks on Cardiorespiratory Parameters at Rest and after Performing the Six-Minute Walk Test in Older Adults"

_geriatrics, 2022, doi:10.3390/geriatrics7030062_

Round 1
Reviewer 1 Report
The study is interesting and simple and could provide information regarding the influence of the mask on cardio-respiratory variables. But the description of the method is very poor. It is not known how it has been measured and the reliability of the meters and instruments.
It is unclear how much time elapsed between the end of the 6MWT and the taking of the measurements. In addition, if the measurements after the 6MWT were made without a mask, the time elapsed between the end of the test and the completion of the measurements may interfere with the results.
Respiratory rate is "breaths per minute", not "beat per minute".
The authors do not indicate how many groups there were, how many subjects in each group, or what the groups are for.
It is not known how they made the measurements, how many observers, which devices.
If there were several observers. Has inter-observer variability been studied?
If there were several devices to measure the variables. Has inter-device variability been studied?
How has dyspnea been assessed?
In the 6MWT they do not describe if the subjects did it individually, in pairs, groups...
Descriptive values of heart rate and respiratory rate should be with figures to a single decimal place.
In material and method, they do not cite or describe "Rate of Perceived Exertion" which is in results.
Line 87: The authors write: "Each older adult performed the 6MWT according to the group to which they had been assigned." They do not describe or analyze any variable by groups.
Lines 155 and 156. The authors write: "... were not significantly different among groups,..."; but we don't know what the groups are. There are no data differentiated by groups.
Line 156. "White dyspnea"?
Lines 178-85. They repeat concepts from the previous paragraph. They don't bring anything new. This paragraph is dispensable.
The authors describe that there are men and women in their population, but do not say if there are differences between the sexes.
They could include this data to improve their results.
In Table 1 we see that 30% of the subjects are hypertensive. Does hypertension influence the results?
In the discussion the authors do not analyze the techniques, procedures and instruments they have used. Nor do they describe the limitations of their work.
The discussion is very poor they repeat the same findings several times.
Conclusions. It's just like the discussion and the results.
References.
There are mistakes. For example
No 1 and 8. The initial letters of each word in the journal name must be capitalized. The names of the journals must be with their abbreviation.
No2. There are too many letters “Jj” in the name of the second author.
No3. Missing (,) after the name of the journal
No 4 and No 5. It is necessary to review the number of authors cited before placing "et al".
No 5. October is not the name of the authors.
The use of bold and italic letters is chaotic.
I make no further corrections. They must be done by the authors.
Reviewer 2 Report
The project presented in the submitted paper is of very low scientific significance. The study group is very small. The division of this group into 6 subgroups is not clearly described. There is no information if the study subjects with arterial hypertension had any possible left ventricle ECHO dysfunction that should have been used as the exclusion criterium. We do not learn anything about a general physical capacity of the examined subjects. Dyspnea is a very subjective sign, and can be also provoked by stressful condition, like wearing any cover over the face. The RPE test, presented finally as the only considerable change, is not even mentioned in the Method subchapter. Because of those many imperfections that rather are impossible to be corrected without major changes in study design, in my opinion the said paper is not suitable to be published in the Geriatrics journal.
Reviewer 3 Report
1. Overall Strengths
The manuscript has not have reach merits. The number of participants is not significance, very short with 40 participants, and poor description of the methodology/statistics. The text is structured and is dificult to follow. there are several language related errors and thus the manuscript requires a thorough checking by a native speaker of English.
2. Importance
The topic is interesting potential to offer more solid evidence based information about the effects of wearing face masks on cardiorespiratory parameters at rest and after performing the six minute walk test in older adults
Hypotheses are needed. What do the authors expect to find based on previous work? This would also be a good place to justify why this study is needed.
3. Justification/Rationale
The justification of this study could be strengthened by explaining what this study adds to the pre-existing literature in the introduction section and comparing with recently researches of De Yñigo-Mojado associated with some variables as the fit factor of masks, the facial hair using masks
Please make it clearer already in the Introduction, what new this study has to offer (i.e. in what way it differs from earlier studies).
3. Methods/Approach
The methodology is not rationale, seriously flawed from the methodological point of view related with the number record type of the design and trial registry before the first participant is enrolled into the study creates a publicly-available record of the researcher’s intentions; including key details such as how many patients they need to recruit, what their outcomes will be, how they intend to measure these outcomes, etc. I appreciate that the authors included a sample size calculation.
Also, Add a study flow chart for the readers ?
4. Results/Findings
The analysis is incomplete in all the tables – what tests were used when data were normally distributed?
The results maybe may be statistically significant, but is it clinically relevant?
Include p-values in all the tables
5. Discussion
Discussion is muddled, confusing to follow and repeats somewhat the Introduction. Furthermore, some more shortcomings should be included, particularly the fact the bias found. The authors could also consider some more ideas for further studies.
6. Conclusions
Write this section part again, clearly and include causative conclusions are warranted.
Round 2
Reviewer 1 Report
This version of the article has improved the manuscript, but something more is needed.
· The authors do not write in a clear way the objective of their study. Now, in the article there is a hypothesis, but it lacks objective.
· Line 123. The original Borg scale contains 15 points, running from 6 to 20, and was developed to measure perceived exertion. The Category Ratio (CR) scale contains 10 points, running from 1 to 10, and was originally developed to measure dyspnea. To assess dyspnea, the CR10 scale should be used and the original should be used to assess the perceived effort.
· In line 175-6 the authors write: "while wearing face masks impacted dyspnea after the 6MWT". It is not correct. As they have used a scale to rate of perceived effort should say “RPE” and not "dyspnea".
· In the tables it is correct. They write "RPE" and do not write dyspnea.
· It the foot of the tables, authors should not remove "beats per minute". "bpm" is "beats per minute" and also "breaths per minute". They must keep both.
· Limitations could go into the discussion. Before the conclusions. If the authors consider that the sample size conditions the significance of the results, they should increase the number of subjects before publishing the results. On the contrary, if they consider that it is enough to ensure their results, it is not a limitation.
· The authors' conclusion is a practical application of their findings., but not the conclusion of the study. The conclusion should be the response to the objective or the acceptance or rejection of the hypothesis. What you write, it is not the conclusion of his work.
· The reference of the data written in line 198 is missing. The new information on lines 184-92 is not analyzed. The authors provide the data of another work but do not relate it to their own or value them. It's not discussion is information.
· In lines 203-7 the authors write about surgical and cloth masks, but not "without a mask". If there are no differences between the three situations, why do they only justify what has happened in two of them? When the subjects do the test "without a mask" they do not have resistance to inspiration and air exhalation but they obtain the same values as when they do the test "with masks". The authors' justification does not seem correct.
I believe that the discussion with respect to the first version has been improved, but it is still not correct. Authors need to improve their arguments.
·
Author Response
Responses to Reviewer #1
- This version of the article has improved the manuscript, but something more is needed. The authors do not write in a clear way the objective of their study. Now, in the article there is a hypothesis, but it lacks objective.
Answer: We would like to thank the reviewer for this suggestion. We have added the objective in the introduction part of the revised manuscript. It now reads “Therefore, this study aims to evaluate the effects of wearing cloth masks and surgical masks on respiratory rate, heart rate, blood pressure, oxygen saturation and perceived exertion at rest and after performing a 6MWT in older adults” (page 2, paragraph 2, lines 63-65).
- Line 123. The original Borg scale contains 15 points, running from 6 to 20, and was developed to measure perceived exertion. The Category Ratio (CR) scale contains 10 points, running from 1 to 10, and was originally developed to measure dyspnea. To assess dyspnea, the CR10 scale should be used and the original should be used to assess the perceived effort.
Answer: We would like to thank the reviewer for this suggestion. We have revised and added new information about the Borg rating of perceived exertion scale. It now reads “Borg rating of perceived exertion scale (6-20 scales) was used to evaluate perceived exertion” (page 3, paragraph 3, lines 124-125). We have revised dyspnea into perceived exertion in the revised manuscript.
- In line 175-6 the authors write: "while wearing face masks impacted dyspnea after the 6MWT". It is not correct. As they have used a scale to rate of perceived effort should say “RPE” and not "dyspnea".
Answer: We would like to thank the reviewer for this suggestion. We have revised dyspnea into perceived exertion in the revised manuscript.
- In the tables it is correct. They write "RPE" and do not write dyspnea.
Answer: We would like to thank the reviewer for this suggestion.
- It the foot of the tables, authors should not remove "beats per minute". "bpm" is "beats per minute" and also "breaths per minute". They must keep both.
Answer: We would like to thank the reviewer for this suggestion. We have added “bpm= beats per minute” at the foot of the tables.
- Limitations could go into the discussion. Before the conclusions. If the authors consider that the sample size conditions the significance of the results, they should increase the number of subjects before publishing the results. On the contrary, if they consider that it is enough to ensure their results, it is not a limitation.
Answer: We would like to thank the reviewer for this suggestion. We have added limitations into the discussion part as suggested by the reviewer. In addition, we deleted the information of sample size from the limitations of this study because we calculated the sample size for this study. Therefore, the sample size was sufficient to ensure our results. It now reads “There were several limitations. First, in this study there was no comparison made between men and women for the different cardiorespiratory parameters and distance of the 6MWT in the six mask conditions. This matter of comparison of the effects of face masks on cardiorespiratory parameters between men and women should be investigated in further studies. Secondly, further studies should evaluate the psychological status of the volunteers, taking into consideration anxiety and stress, since these conditions can increase dyspnea when wearing a facemask, and can be a source of bias in the results” (page 10, paragraph 1, lines 238-245).
- The authors' conclusion is a practical application of their findings., but not the conclusion of the study. The conclusion should be the response to the objective or the acceptance or rejection of the hypothesis. What you write, it is not the conclusion of his work.
Answer: We would like to thank the reviewer for this suggestion. We have revised the conclusion section of the revised manuscript as suggested by the reviewer. It now reads “cloth masks and surgical masks had no impact on respiratory rate, heart rate, blood pressure, oxygen saturation at rest and after performing the 6MWT, but had an impact on perceived exertion after the 6MWT in older adults” (page 10, paragraph 2, lines 255-257).
- The reference of the data written in line 198 is missing. The new information on lines 184-92 is not analyzed. The authors provide the data of another work but do not relate it to their own or value them. It's not discussion is information.
Answer: We would like to thank the reviewer for this suggestion. We have added the reference of the data written in line 198. We have adjusted the information on lines 184-192 of the revised manuscript as suggested by the reviewer. It now reads “The authors tested cloth masks and surgical masks in 50 hospital employees (median age 33 years; 32% with comorbidities) while sitting for 10 minutes. The results showed that sitting while wearing a cloth mask and a surgical mask brought about no statistically significant differences in heart rate and oxygen saturation when compared to the baseline. In addition, another previous study investigated surgical masks at 5 and 30 minutes of rest in 15 healthy volunteers (average age 31.1±1.9 years) and 15 veterans with severe chronic obstructive pulmonary disease (COPD) (average age 71.6 ±8.7years). Those results found no major change in oxygen saturation at 5 and 30 minutes of rest in healthy volunteers and those with severe COPD. Furthermore, the authors did not compare oxygen saturation between two populations [22]. These findings suggested that cloth masks and surgical masks did not impact the respiratory rate, heart rate, blood pressure, and oxygen saturation at rest for less than 30 minutes in individuals with or without underlying pulmonary disease [22]. Our study tested the three conditions of face masks in 40 older adults without pulmonary disease sitting for 10 minutes. The results showed that no differences in respiratory rate, heart rate, blood pressure and oxygen saturation were recorded among the conditions. Therefore, further studies may consider longer time periods and compare cardiorespiratory parameters between healthy volunteers and those with pulmonary disease” (page 8, paragraph 3, lines 184-205).
- In lines 203-7 the authors write about surgical and cloth masks, but not "without a mask". If there are no differences between the three situations, why do they only justify what has happened in two of them? When the subjects do the test "without a mask" they do not have resistance to inspiration and air exhalation but they obtain the same values as when they do the test "with masks". The authors' justification does not seem correct.
Answer: We would like to thank the reviewer for this suggestion. We have adjusted this information of the revised manuscript as suggested by the reviewer. It now reads “After performing the 6MWT, respiratory rate, heart rate and blood pressure slightly increased, while oxygen saturation slightly decreased. This change occurred because the prolonged inspiratory activity leads to high negative intrathoracic pressure, leading to increased cardiac preload, systolic volume and cardiac afterload, respectively [23, 24].” (page 9, paragraph 2, lines 208-212).
- I believe that the discussion with respect to the first version has been improved, but it is still not correct. Authors need to improve their arguments.
Answer: We would like to thank the reviewer for this useful suggestion. We have adjusted the discussion section of the revised manuscript as suggested by the reviewer. It now reads “The study findings determined that cloth masks and surgical masks had no impact on cardiorespiratory parameters including respiratory rate, heart rate, blood pressure, oxygen saturation at rest and after performing the 6MWT, while wearing face masks impacted perceived exertion after the 6MWT. The cloth masks and surgical masks did not impact respiratory rate, heart rate, blood pressure, and oxygen saturation after sitting for 10 minutes. These findings concurred with a previous study [21]. The authors tested cloth masks and surgical masks in 50 hospital employees (median age 33 years; 32% with comorbidities) while sitting for 10 minutes. The results showed that sitting while wearing a cloth mask and a surgical mask brought about no statistically significant differences in heart rate and oxygen saturation when compared to the baseline. In addition, another previous study investigated surgical masks at 5 and 30 minutes of rest in 15 healthy volunteers (average age 31.1±1.9 years) and 15 veterans with severe chronic obstructive pulmonary disease (COPD) (average age 71.6 ±8.7years). Those results found no major change in oxygen saturation at 5 and 30 minutes of rest in healthy volunteers and those with severe COPD. Furthermore, the authors did not compare oxygen saturation between two populations [22]. These findings suggested that cloth masks and surgical masks did not impact the respiratory rate, heart rate, blood pressure, and oxygen saturation at rest for less than 30 minutes in individuals with or without underlying pulmonary disease [22]. Our study tested the three conditions of face masks in 40 older adults without pulmonary disease while sitting for 10 minutes. The results showed no differences in respiratory rate, heart rate, blood pressure and oxygen saturation were recorded among the conditions. Therefore, further studies may consider longer time periods and compare cardiorespiratory parameters between healthy volunteers and those with pulmonary disease. After performing the 6MWT, respiratory rate, heart rate and blood pressure slightly increased, while oxygen saturation slightly decreased. This change occurred because the prolonged inspiratory activity leads to high negative intrathoracic pressure, leading to increased cardiac preload, systolic volume and cardiac afterload, respectively [23, 24]. This gave an increased respiratory rate, heart rate, and blood pressure. Decreased oxygen saturation reflected reduced oxygen levels in the blood. These results concurred with previous studies showing that healthy subjects with no mask and surgical masks had no statistically significant differences in the respiratory rate, heart rate and oxygen saturation after performing the 6MWT [12, 25]. In contrast, our results found that wearing cloth masks and surgical masks statistically significantly increased dyspnea when compared with no masks. The face masks increased airflow resistance, facial skin temperature and moisture or heat of the inhaled air, resulting in increased dyspnea [26, 28]. Previous studies reported that healthy subjects wearing face masks recorded similar distances walked during the 6MWT compared to no masks [25, 29, 30], while changes after the 6MWT were found in heart failure or chronic lung disease patients [31, 32]. This study involved no participants with heart failure or chronic lung disease.
There were several limitations. First, in this study there was no comparison made between men and women for the different cardiorespiratory parameters and distance of the 6MWT in the six mask conditions. This matter of comparison of the effects of face masks on cardiorespiratory parameters between men and women should be investigated in further studies. Secondly, further studies should evaluate the psychological status of the volunteers, taking into consideration anxiety and stress, since these conditions can increase dyspnea when wearing a facemask, and can be a source of bias in the results” (page 8-13, paragraph 2, lines 173-245).
Reviewer 2 Report
Because all my remarks have been taken into considerations and the suitable improvements and corrections were introduced to the manuscript text accordingly, therefore I recommend this paper for publication in the Geriatrics.
Author Response
Responses to Reviewer #2
- Because all my remarks have been taken into considerations and the suitable improvements and corrections were introduced to the manuscript text accordingly, therefore I recommend this paper for publication in the Geriatrics.
Answer: We would like to thank the reviewer for consideration of this manuscript.
Reviewer 3 Report
The authors have clearly and adequately addressed all comments raised by the reviewers. Please also consider to include the section seven "limitations" in the discussion section.
Author Response
Responses to Reviewer #3
- The authors have clearly and adequately addressed all comments raised by the reviewers. Please also consider to include the section seven "limitations" in the discussion section.
Answer: We would like to thank the reviewer for this suggestion. We have added the section of limitations into the discussion section of the revised manuscript. It now reads “There were several limitations. First, in this study there was no comparison made between men and women for the different cardiorespiratory parameters and distance of the 6MWT in the six mask conditions. This matter of comparison of the effects of face masks on cardiorespiratory parameters between men and women should be investigated in further studies. Secondly, further studies should evaluate the psychological status of the volunteers, taking into consideration anxiety and stress, since these conditions can increase dyspnea when wearing a facemask, and can be a source of bias in the results” (page 10, paragraph 1, lines 238-245).